# Identification of the cleavage sites leading to the shed forms of human and mouse anti-aging and cognition-enhancing protein Klotho

**Ci-Di Chen[1], Yuexuan Li[1]⦾, Arthur K. Chen[1]⦾, Melissa A. Rudy[1]⦾, Jason S. Nasse[1], Ella Zeldich[1], Taryn J. Polanco[1], Carmela R. Abraham[1,2]***

**1** Department of Biochemistry, Boston University School of Medicine, Boston, Massachusetts, United States of America, **2** Department of Pharmacology and Experimental Therapeutics, Boston University School of Medicine, Boston, Massachusetts, United States of America

⦾ These authors contributed equally to this work.

\* cabraham@bu.edu

**Data Availability Statement:** All relevant data are within the paper and its Supporting Information files.

## Abstract

Klotho is an age-extending, cognition-enhancing protein found to be down-regulated in aged mammals when age-related diseases start to appear. Low levels of Klotho occur in neurodegenerative diseases, kidney disease and many cancers. Many normal and pathologic processes involve the proteolytic shedding of membrane proteins. Transmembrane (TM) Klotho contains two homologous domains, KL1 and KL2 with homology to glycosidases. After shedding by ADAM 10 and 17, a shed Klotho isoform is released into serum and urine by the kidney, and into the CSF by the choroid plexus. We previously reported that human Klotho contains two major cleavage sites. However, the exact cleavage site responsible for the cleavage between the KL1 and KL2 domains remains unknown for the human Klotho, and both sites are unknown for mouse Klotho. In this study, we aimed to identify the cleavage sites leading to the shed forms of human and mouse Klotho. Mutations in the region close to the TM domain of mouse Klotho result in the reduced shedding of the 130 kD (KL1 +KL2) and 70 kD (KL1) fragments, suggesting that the cleavage site lies within the mutated region. We further identified the cleavage sites responsible for the cleavage between KL1 and KL2 of human and mouse Klotho. Moreover, mutated Klotho proteins have similar subcellular localization patterns as wild type Klotho. Finally, in an FGF23 functional assay, all Klotho mutants with a nine amino acid deletion can also function as an FGFR1 co-receptor for FGF23 signaling, however, the signaling activity was greatly reduced. The study provides new and important information on Klotho shedding, and paves the way for studies aimed to distinguish between the distinct roles of the various isoforms of Klotho.

## Introduction

As the population, especially in developed contries, lives longer due to medical advances, so do we see an increase in age-related diseases affecting the nervous system. Among them are a number of neurodegenerative diseases such as Alzheimer's (AD), Parkinson's and

**Funding:** This study was supported in part by an NIH-NIA grant R56 AG051638 to CRA and by an Affinity Research Collaborative grant from the Boston Medical Center Evans Foundation to CRA. https://www.bumc.bu.edu/evanscenteribr/. The funders had no role in study design, data collection and analysis, decision to publish, or preparation of the manuscript. There was no additional external funding received for this study.

**Competing interests:** The authors have declared that no competing interests exist.

amyotrophic lateral sclerosis (ALS). Thus, any remedy that could prevent, delay, alleviate or even cure these brain disorders is ugently needed. Even in the absence of disease, a decline in cognitive function is observed in the aging individuals. The neurodegenative diseases, and, to a lesser extent, normal aging share a number of characteristics including oxidative stress, inflammation, reduced myelin integrity and a decreased level of neuroprotection which is manifested by the resilience of the brain to various insults.

We reported that the expression of Klotho, an anti-aging and cognition-enhancing protein is markedly downregulated in the brains of aged monkeys, mice and rats [1]. Klotho is highly expressed in the kidney and, to a lesser extent, in the brain and reproductive organs. In the brain, Klotho is expressed mainly in the ependymal cells of the choroid plexus, in hippocampal neurons, cerebellar Purkinje cells and also is detected in cerebral white matter [2]. Mice deficient in Klotho exhibit changes observed during human aging [3] including arteriosclerosis, osteoporosis, cognitive decline and neurodegeneration of hippocampal and cerebellar and Purkinje neurons [3]. They also have a shorten lifespan. In contrast, mice overexpressing Klotho live 30% longer than wild type mice [3, 4]. We pioneered the exploration of Klotho's functions in the brain. Impaired cognition is observed in Klotho-deficient mice [5], while enhanced Klotho expression reduces cognitive deficits in a mouse model of AD [6] and improves cognitive functions in young [7] and old mice [8]. Recent work on Klotho's functions in the CNS, performed by us and by others, demonstrated that Klotho can combat processes associated with neurodegeneration through diverse mechanisms. Klotho fosters survival of hippocampal [9, 10], dopaminergic [11, 12] and cortical neurons [13] and cerebellar granule cells; and counteracts oxidative stress [4]. Klotho enhances oligodendrocyte differentiation and maturation *in vitro* [14], and enhances re-myelination *in vivo* in the cuprizone-induced demyelination model of multiple sclerosis (MS) [15]. Klotho levels are decreased in the CSF of AD [16] and MS patients [17], supporting its role in neuroprotection and myelin health. Genetic studies identified a Klotho polymorphism (KL-VS) associated with higher serum levels of Klotho. KL-VS haplotype has been associated with longevity [18, 19], enhanced cognition [7] and greater frontal brain volume and executive function across all examined ages [20].

Klotho exists in three forms, each with its own unique functions: full-length transmembrane Klotho (FL-KL), shed Klotho (shKL), and secreted Klotho (sKL) which is produced via differential splicing [21]. In the kidney, FL-KL and shKL function as a co-receptor with FGFR1 for FGF23 signaling, which regulates Vitamin D levels in the serum. The shed Klotho acts as a hormone on remote tissues. Several papers reported that shKL acts as a sialidase on ion channels for Klotho's other functions such as ion homeostasis [22, 23] However, a recently published study exposed the crystal structure of Klotho and determined that shKL acts as a non-enzymatic scaffold that concominantly links FGF23 with FGFR hense stabilizing the interaction between the two and promoting the downstream signaling [24]. In other tissues, Klotho plays a role in anti-inflammation, tumor suppression, senescence, cell differentiation, and cardiovascular function [25–27].

The shedding of membrane proteins plays a major role in development, inflammation, and disease. Klotho is shed from the cell surface by alpha-secretases ADAM10 and ADAM17 [28], but also by beta and gamma secretase [29]. Two major cleavage sites in Klotho that are recognized by ADAM10 and ADAM17 were previously identified: one close to the juxtamembrane region (alpha1) and the second between the KL1 and KL2 domains (alpha2) [30].

However, the precise site (alpha2) responsible for the cleavage between KL1 and KL2 remains unknown for the human Klotho and both sites (alpha1 and alpha2) are unknown for mouse Klotho. In this study, we aimed to identify the cleavage sites leading to the shed form of human and mouse Klotho by mutating potential sheddase recognition sequences, transfecting

the mutants into HEK-293 cells, and examining the presence of Klotho's extracellular 130 kD (KL1+KL2) and 70 kD (KL1) fragments in cell lysates and media.

## Materials and methods

### Mutagenesis and plasmids construction

The mutants used in this study are listed in Fig 1D. The mutations were introduced into Klotho cDNA in pcDNA3.1 vector and KL-V5 plasmid [28] using the In-Fusion Cloning kit (Clontech, Mountain View, CA) with the following sense and antisense primers, respectively: hKLΔ9L4b mutant: 5′–TTCGGCGGCTCGGGCGAAGGGACATTTCCCTGTGAC–3′ and 5′–GCCCGAGCCGCCGAAGCCATTTTTCTCTATCAGC–3′; and hKLΔ9L9b mutant: 5′–GGCGGCTCGGGCGGCGAAGGGACATTTCCCTGTGAC–3′ and 5′–GCCGCCCGAGCC GCCGCCCGAGCCGCCGAAGCC–3′. To construct hKLΔ9L4b and hKLΔ9L9b tagged with V5 at the C-terminus, the EcoRI—AgeI fragment from KL-V5 [28] was ligated into the hKLΔ9L4b and hKLΔ9L9b plasmids, respectively. To construct hKLΔ9L4b and hKLΔ9L9b tagged with GFP at the C-terminus, a similar technique was used: the EcoRI—eI fragment from KL-GFP [28] was ligated into the hKLΔ9L4b and hKLΔ9L9b plasmids, respectively.

The mouse KL (mKL) mutations were constructed with the following sense and antisense primers, respectively: mKLΔ9 mutant: 5′–GACAGCAATGGCTTCTGTCCAGAAGAATA CACTGTGTGC–3′ and 5′–GAAGCCATTGCTGTCAATAATTTTCC–3′; mKLΔ9b mutant: 5′–GAGGACAATGGCTTTGAAGGGACATTTCCCTGTGAC–3′ and 5′–AAAGC CATTGTCCTCTATCAGC–3′; mKLΔ9L4 mutant 5′–TTCGGAGGATCCGGATGTCCAGAA GAATACACTGTGTGC–3′ and 5′–TCCGGATCCTCCGAAGCCATTGCTGTCAATAATTT TCC–3′: mKLΔ9bL4 mutant 5′–TTTGGAGGATCCGGAGAAGGGACATTTCCCTGTGAC–3′ and 5′–TCCGGATCCTCCAAAGCCATTGTCCTCTATCAGC–3′: and mKLΔ9L9 mutant: 5′–TTCGGAGGATCCGGAGGAGGATCCGGAGGATGTCCAGAAGAATACACTGTGTGC–3′ and 5′–TCCGGATCCTCCGAAGCCATTGCTGTCAATAATTTTCCTG–3′; mKLΔ9bL9 mutant: 5′–TTTGGAGGATCCGGAGGAGGATCCGGAGGAGAAGGGACATTTCCCTGTGACT–3 and' 5′–TCCGGATCCTCCAAAGCCATTGTCCTCTATCAGCT–3′.

To construct mKL, mKLΔ9 and mKLΔ9b tagged with myc tag at the C-terminus, a similar technique was used. Sense and antisense primers are: 5′– TCTAGAGGGCCCTTCGAA CAAAAAC –3 and' 5′– GAAGGGCCCTCTAGACTTATAACTTCTCTGGCCTTTCTTGGAG–3′. All plasmids were confirmed by DNA sequencing.

### Cell maintenance

HEK-293 and COS-7 cells were maintained in Dulbecco's modified Eagle's medium (DMEM) (CellGRo) supplemented with 10% fetal bovine serum (FBS) (Atlantic Biologicals), and 1% penicillin-streptomycin (CellGro) at 37°C and 5% $CO_2$. The cells were split twice a week until the experiment was started.

### Western blotting

Details of Western Blotting were described previously [31].

The rat monoclonal anti-KL antibody KM2076 (1:2000) was described in [32] and was purchased from TransGenic Inc, (Tokyo, Japan). The antibodies to total ERK and p-ERK were from the phospho ERK pathway kit (Cell Signaling, Danvers, MA) and were used according to the manufacturer's protocol. The mouse anti-V5 antibody (1:5,000) was from Invitrogen (NY, NY).

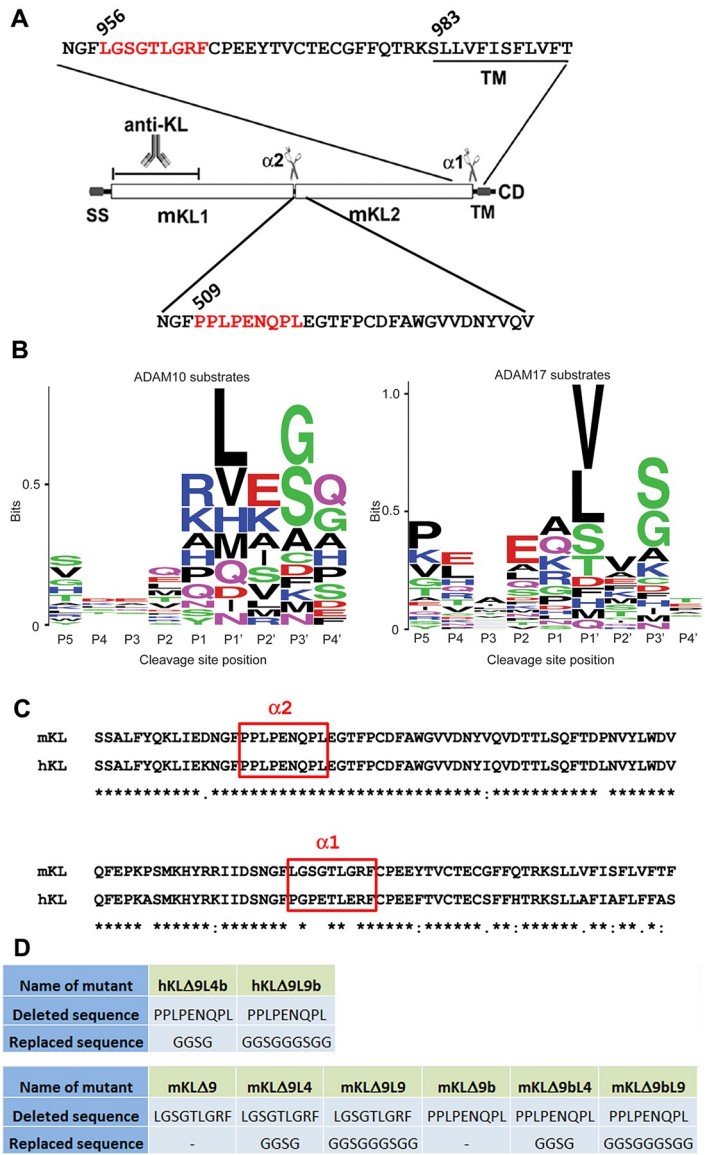

**Fig 1. Predicted ADAM10 and ADAM17 cleavage sites in the Klotho protein.** A) Diagram of the mouse Klotho protein structure with an enlarged view of the two potential cleavage sites. The nine amino acids that were deleted to form the mutants are marked in a red box. B) Representation of highly conserved residues found at cleavage sites in 34 known ADAM10 and ADAM17 substrate proteins [30] were compiled and used to generate a sequence logo with the program WebLogo 3.0. Logos represent preferred amino acid residues for each primed and unprimed position. The height of the letter is proportional to the log of its frequency in the sequence alignment. C) Alignment of mouse Klotho and human Klotho sheddases recognition sites. The two cleavage sites (alpha1 and alpha2) are marked in red boxes. D) Mutants used in this study. The two cleavage sites sequence and the replaced linker sequence are listed.

## Immunofluorescence

COS-7 cells were transfected with various Klotho constructs for 48 hours, fixed and studied via indirect immunofluorescence as described [30].

## Cell-surface biotinylation assay

Cell-Surface Biotinylation Assay was performed as described [30].

### FGF23 signaling assay

FGF23 Signaling Assay was performed as described [30].

## Results

### Determination of potential shedding sites in Klotho

Klotho contains a signal sequence (SS), two homologous domains (KL1 and KL2), a transmembrane domain (TM) and a short cytoplasmic domain (CD). Two cleavage sites for the sheddases ADAM10 and ADAM17 have been previously identified: one close to the transmembrane (TM) region, and another one between the KL1 and KL2 domains (Fig 1A).

However, the exact cleavage site responsible for the cleavage between KL1 and KL2 remains unknown for the human KL and both sites are unknown for the mouse Klotho. Highly conserved residues found at cleavage sites in 34 reported substrate proteins compiled by Caescu et al [33] were examined to predict the ADAM10 and ADAM17 recognition sites in Klotho. Sequence logos were generated from the sequence alignments of the cleavage sites that show the preferred amino acid residues for each primed and unprimed position (Fig 1B). At the P1' position, valine and leucine are among the most frequent residues at ADAM10 and ADAM17 cleavage sites. The predicted recognition patterns and the sequence alignment of human and mouse Klotho suggest the potential ADAM10 and ADAM17 cleavage site close to the TM region is within the LGSGT-LGRF site for the one close to the juxtamembrane region (alpha1) for mouse Klotho and PPLPE-NQPL between the KL1 and KL2 domains (alpha2) for both human and mouse Klotho (Fig 1A).

To determine whether the predicted cleavage site is indeed the sheddases' recognition site, the potential alpha2 site of human Klotho was mutated, replaced with a 4 or 9 amino acid linker, and transfected into COS-7 cells (Fig 2A). For image of the entire blot see S1A Fig. COS-7 and HEK cells express both ADAM10 and ADAM17 sheddases [28, 30]. A deletion of nine amino acids from the alpha2 sites with a four-amino acid linker (KLΔ9L4b) almost completely abolished the 70 kD fragment in the cell lysate, media, and on the membrane (Fig 2A), suggesting the predicted alpha2 site is the sheddases' recognition site. When KLΔ9L4b is expressed, the 130 kD band in the media showed a significant decrease compared to wild type, suggesting the cleavage at the alpha2 site affects cleavage at the alpha1 site. This result confirms a previous report that the cleavage at both sites occurrs simultaneously [30], suggesting that the cleavage of each site may affect each other's cleavage. The mutant with nine-amino acid linker replacing the predicted nine amino acids was not stable and barely detectable.

A similar experiment was performed for mouse Klotho, for which both alpha1 and alpha2 sites were unknown. Mutants included a nine amino acid deletion from the alpha1 site (mKLΔ9), a nine amino acid deletion from the alpha1 site with a nine amino acid linker (mKLΔ9L9), and a nine amino acid deletion from the alpha2 site (mKLΔ9b). Deletion of nine-amino acids at the alpha1 site (mKLΔ9) resulted in a large reduction in the 130 kD fragment in the medium (Fig 2B), suggesting that the predicted site is the sheddase recognition site. For image of the entire blot see S1B Fig. Mutation in the alpha1 site (mKLΔ9) resulted in a reduction in the 70 kD fragment, suggesting that the cleavage at the alpha2 site is dependent on the cleavage at the alpha1 site (Fig 2B). We reported a similar observation for the human KL where the cleavage at the alpha2 site is dependent on the cleavage at the alpha1 site [30]. The mutant with the nine-amino acid linker replacing the predicted nine amino acids (mKLΔ9L9) was not stable and barely detectable (Fig 2B). Deletion of nine-amino acids at the alpha2 site (mKLΔ9b) showed a significant reduction in the 70 kD fragment in the cell lysate, medium,

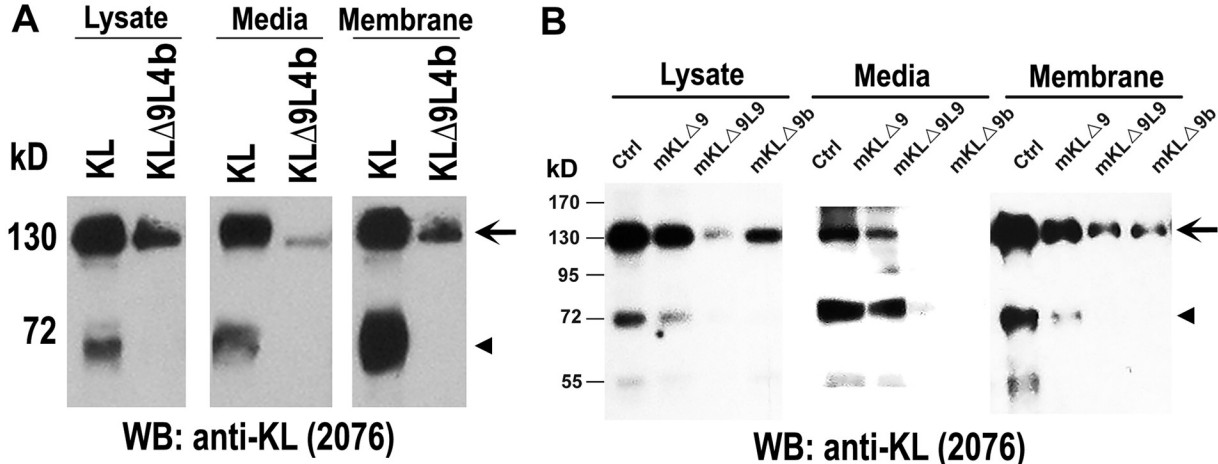

**Fig 2. Determination of potential shedding sites in Klotho.** A) Western Blot of indicated human Klotho plasmids transfected in COS-7 cells. Protein samples were collected from either cell lysates, media or after cell-surface biotinylation assay (membrane proteins). Arrow indicates that the KLΔ9L4b 70 kD band is lost compared to KL-WT. B) Western Blot of indicated mouse Klotho plasmids transfected in HEK cells. Protein samples were collected from either cell lysates, media or after cell-surface biotinylation assay (membrane proteins). Arrow indicates the 130 kD protein and arrowhead the 70 kD Klotho fragment.

and membrane as compared to wild type mKL, indicating that the recognition sequence most likely lies within the mutated region, as predicted.

Similarly, when mKLΔ9b is expressed, the 130 kD band in the medium is undetectable compared to wild type (Fig 2B), suggesting that the cleavage at the alpha2 site affects cleavage at the alpha1 site.

## Subcellular localization of the Klotho mutant

Immunofluorescence was performed to determine whether the mutations in Klotho affected its subcellular localization. COS-7 cells were transfected with mKL-WT, mKLΔ9, mKLΔ9b, or either GFP- or V5-tagged hKL-WT and hKLΔ9L4b. The results show that all mutants distribute normally and similarly to KL WT (Fig 3).

## Functional assay of the Klotho mutant

An FGF23 signaling assay was performed to determine whether the deletion of nine amino acids altered the function of Klotho. HEK cells were transfected with FGFR1 and either hKL or hKLΔ9L4b (mutation in alpha2 site). Signaling was measured via ERK phosphorylation with Klotho acting as a co-receptor. The results show that after 20 minutes of incubation with 10 ng/mL FGF23, the KLΔ9L4b mutant preserves some function for FGF23 signaling, however, it exhibits a large reduction in ERK phosphorylation as compared to wild type Klotho (Fig 4A). For image of the entire blot see S2A Fig. This indicates the mutation modifies human Klotho's ability to act as a co-receptor with FGFR1.

A similar experiment was performed to study the effect different doses of FGF23 have on ERK phosphorylation at the alpha2 site of mutant human Klotho and mouse mutants with mutations in both the alpha1 and alpha2 sites. The results show that mouse alpha1 mutant (mKLΔ9) only exhibited about 50% ERK phosphorylation compared to Klotho WT (Fig 4B). For image of the entire blots see S2Ba–S2Be Fig. The results suggest that mKLΔ9 mutant has comparable function as a FGFR1 co-receptor as KL WT on the cell surface, and this is

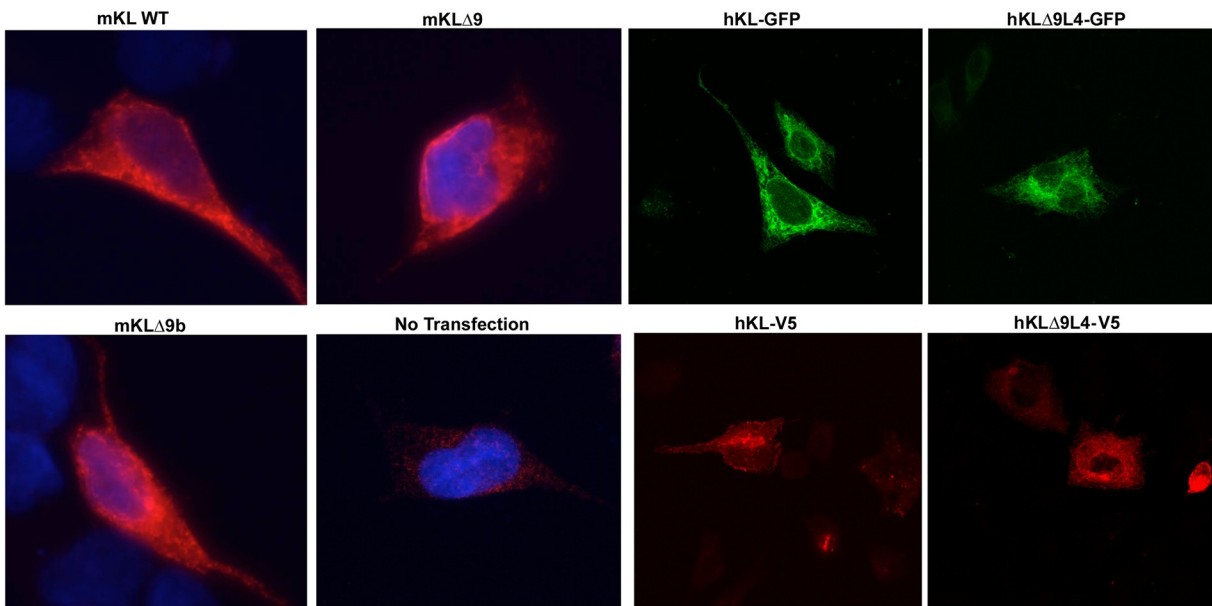

**Fig 3. Klotho mutants have similar sub-cellular localization which resembles WT KL distribution.** Indirect immunofluorescence shows the sub-cellular localization of Klotho mutants compared to WT KL. COS-7 cells were transfected with indicated plasmids for 48 hours, fixed and studied using either anti-KL (KM2076) or anti-V5 antibodies that recognizes the transfected Klotho-V5 protein.

consistent with the results we found with the human alpha1 mutants [30]. Both human and mouse alpha2 mutants showed reduced ERK phosphorylation abilities compared to KL WT and mKLΔ9 mutant (Fig 4B). This indicates that a mutation at the alpha2 site modifies human Klotho's ability to act as a co-receptor with FGFR1, likely because this region does not simply function as a flexible linker, but is more critical for Klotho's structure-function, and a mutation in this region results in reduced activity. Nevertheless, both alpha2 mutants retain some activity as co-receptors with FGFR1(Fig 4B).

## Discussion

Analysis of peptide substrate libraries indicates that ADAM10 and ADAM17 have strong sequence preferences, yet there is no particular residue that is absolutely conserved at a given position in the known cleavage sites. The information from the peptide library screening was used to predict and identify the ADAM10 and ADAM17 recognition sites at the alpha1 site (close to the TM region) as LGSGTLGRF (Fig 1A). For the alpha2 site, the recognition amino acid sequences are PPLPENQPL for both mouse and human Klotho. Deletion of these amino acids results in a reduction of protein cleavage.

The crystal structure of Klotho is available (PDB access number: 5W21_A, [24]. Based on NCBI's conserved domain database for the functional annotation of proteins [34], Klotho structure contains two domains (KL1 and KL2) that belong to Glyco_hydro_1 Superfamily (access number: cl23725). KL1 domain ranges from aa 58 to 506, and KL2 domain ranges from aa 517 to 953, leaving a 10 amino acid linker region of PPLPENQPLE. The recognition sequence we identified is exactly within the linker region predicted by NCBI conserved domain search.

There are two major cleavage sites in the Klotho protein that are recognized by sheddases: one close to the TM region (alpha1) and one in between the KL1 and KL2 domains (alpha2)

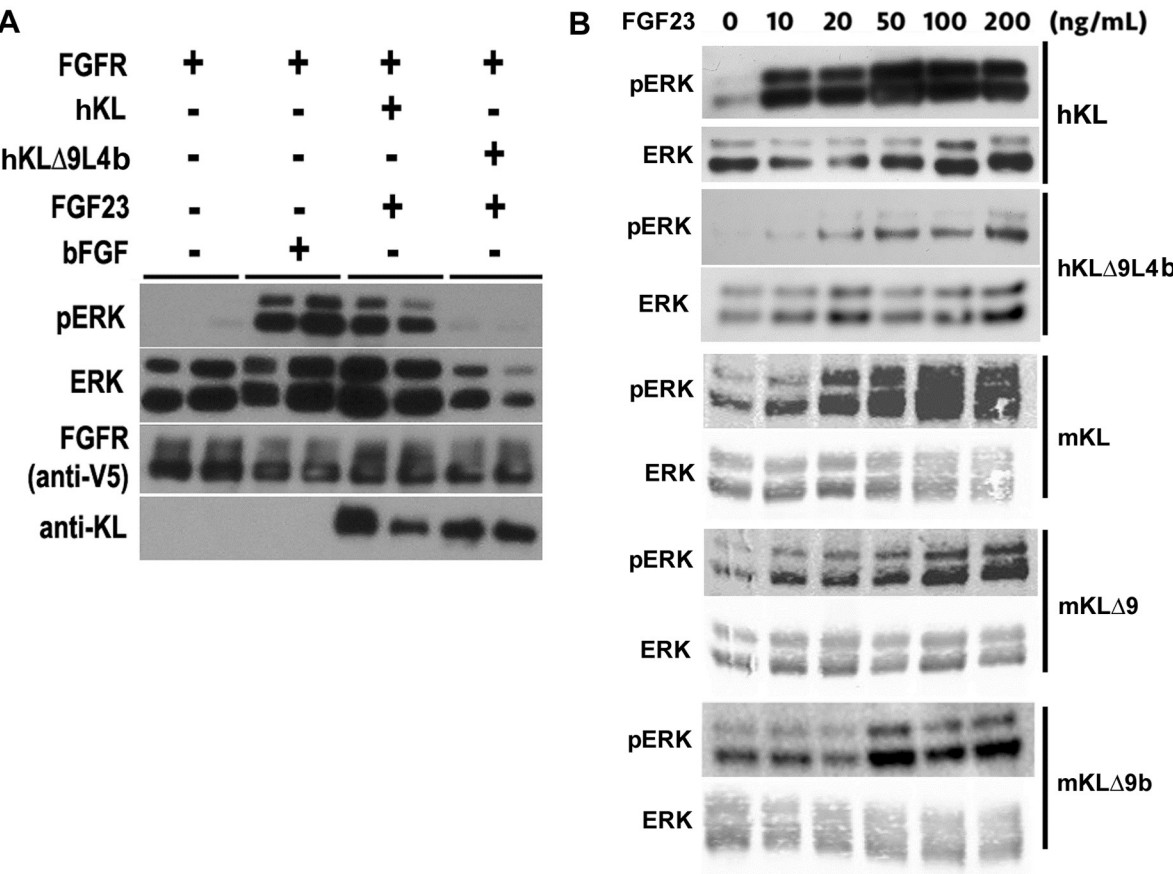

**Fig 4. FGF23 signaling activity of Klotho mutants.** A. Western Blot showing differences in ERK phosphorylation compared to total expression of ERK after transfection of the V5-tagged FGF receptor and KL WT or KLΔ9L4b. Cells were treated with 10 ng/mL FGF23 for 30 minutes. bFGF is used as positive control for ERK phosphorylation. Arrow marks the reduction in the pERK doublet by KLΔ9L4b relative to KL WT. B. High concentrations of FGF23 increases ERK phosphorylation in mouse and human KL mutants. Western Blot of HEK-293 cells transfected with human and mouse Klotho mutants with different doses of FGF23 for 15 minutes. Western Blotting was performed with either pERK or total ERK antibodies.

(Fig 1A). The deletion of the amino acids LGSGTLGRF from the alpha1 site in mouse KL resulted in a reduction of both the 130 kD and 70 kD fragments of the protein (Fig 2B and 2C). This suggests that the cleavage that takes place between the KL1 and KL2 domains is dependent upon cleavage near the TM domain.

There is a major clinical and scientific significance in understanding of the processing and cleavage of Klotho. For example, Klotho is known to be essential to act as a co-receptor for FGF23 in order to activate FGFR, and the altered expression of either Klotho or FGF23 could result in higher than normal levels of Vitamin D, 1 alpha-hydroxylase and 1,25(OH)2D3 [35, 36]. The discrepancy of this observation is in the fact that Klotho is predominantly produced in the distal convoluted tubules in the kidney, while vitamin D synthesis as well as the reabsorption of phosphate occurs mainly in the proximal tubules. One of the explanations for the activation of the FGF signaling cascade by FGF23 in proximal tubules in the absence of locally expressed Klotho is that Klotho, shed from the distal convoluted tubules, acts as a paracrine factor and activates FGFR on proximal tubules to facilitate the response to FGF23 [37]. Therefore, understanding the mechanism and the factors controling the shedding of Klotho in different tissues can result in potential therapeutics for kidney disease.

The processing of Klotho and its isoforms was demonstrated to be tissue specific. A recent study demonstrated that in the brain the levels of the TM form of Klotho are ten times lower than the levels of Klotho transcript produced by alternative splicing (sKL; 70 kDa) [38] while in the kidney the levels of TM Klotho are much higher than sKL. These findings suggest that different Klotho isoforms serve different functions and that the mechanism of Klotho shedding is differentially regulated in various tissues. Therefore, understanding the mechanisms and the factors controling the shedding of Klotho in different tissues can contribute to the development of therapeutic strategies aimed to increase Klotho levels.

The nine amino acid deletion near the TM resulted in a large reduction in shedding activity. These findings are similar to the results obtained from human Klotho experiments involving a nine amino acid deletion at the alpha1 site [30]. The possible reasons that complete inhibition of shedding could not be achieved may be attributed to the fact that ADAM10 and ADAM17 do not have stringent recognition sequence requirements and they may cleave at near by sites, or sheddases other than ADAM10 and ADAM17 play a role in the proteolytic cleavage activity. Interestingly, in this study, we found that in both human and mouse Klotho, mutation in the alpha2 region affects alpha1 cleavage (Fig 2A and 2B). We found previously that the 130 kD and 70 kD fragments appeared at the same time [30], suggesting that both alpha1 and alpha2 cuts occur simultaneously, and that one cut is dependent on the other. Of course, we can not rule out that mutation at alpha2 site resulted in a protein conformation change that affects the accessibility of the alpha1 cleavage. A functional assay assessing ERK phosphorylation tested the mutants' ability to function as a co-receptor with FGFR1 for FGF23 signaling. The results indicate the mutation largely reduces mouse and human Klotho's ability to function as a co-receptor with FGFR1, although mouse Klotho is affected to a lesser extent.

Full-length KL (FL-KL) and shed KL (shKL) have distinct functions. In future studies we plan on generating knock-in mice expressing the uncleavable KLΔ9 mutant to distinguish between the distinct roles of the two forms of KL. The KLΔ9 knock-in mice line will lack shKL which will allow us to analyze *in vivo* the role shKL plays in neuroprotection in Alzheimer's disease and other neurodegenerative diseases, in myelination and tumor suppression.

## Supporting information

**S1 Fig. The images of the entire blots were used to generate Fig 2.** S1A The figure on the left was used to generate Fig 2A. S1B The middle figures were used to generate Fig 2B.
(PDF)

**S2 Fig. The images of the entire blots were used to generate Fig 4.** S2A image was used to generate Fig 4A. S2B a, b, c. pERK and ERK blots for hKL. Only one of the repeats was used for the publication. S2B d, e The images were used for pERK for mKL and mKLD9. For all other ERK blots, the uncropped images were not saved because the cropped blot had the only visible bands.
(PDF)

## Author Contributions

**Conceptualization:** Ci-Di Chen, Carmela R. Abraham.

**Data curation:** Ci-Di Chen, Yuexuan Li, Arthur K. Chen, Melissa A. Rudy, Jason S. Nasse, Ella Zeldich, Taryn J. Polanco.

**Formal analysis:** Ci-Di Chen, Ella Zeldich, Carmela R. Abraham.

**Funding acquisition:** Carmela R. Abraham.

**Investigation:** Ci-Di Chen.

**Methodology:** Ci-Di Chen.

**Project administration:** Carmela R. Abraham.

**Supervision:** Carmela R. Abraham.

**Writing – original draft:** Ci-Di Chen.

**Writing – review & editing:** Ci-Di Chen, Carmela R. Abraham.

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
