## [Decision Letter · Decision Letter 0]

7 Nov 2019

PONE-D-19-22466

Identification of the cleavage sites leading to the shed forms of human and mouse anti-aging and cognition-enhancing protein Klotho

PLOS ONE

Dear Dr. Abraham,

Thank you for submitting your manuscript to PLOS ONE. After careful consideration, we feel that it has merit but does not fully meet PLOS ONE’s publication criteria as it currently stands. Therefore, we invite you to submit a revised version of the manuscript that addresses the points raised during the review process.

We would appreciate receiving your revised manuscript by Dec 22 2019 11:59PM. To enhance the reproducibility of your results, we recommend that if applicable you deposit your laboratory protocols in protocols.io, where a protocol can be assigned its own identifier (DOI) such that it can be cited independently in the future. For instructions see: http://journals.plos.org/plosone/s/submission-guidelines#loc-laboratory-protocols

We look forward to receiving your revised manuscript.

Kind regards,

Hiroyoshi Ariga

Academic Editor

PLOS ONE

Journal Requirements:

"This study was supported in part by an NIH-NIA grant R56 AG051638 to CRA and by an Affinity Research Collaborative grant from the Boston Medical Center Evans Foundation to CRA. https://www.bumc.bu.edu/evanscenteribr/

The funders had no role in study design, data collection and analysis, decision to publish, or preparation of the manuscript.".

i) Please provide an amended statement that declares *all* the funding or sources of support (whether external or internal to your organization) received during this study, as detailed online in our guide for authors at http://journals.plos.org/plosone/s/submit-now.  Please also include the statement “There was no additional external funding received for this study.” in your updated Funding Statement.

ii) Please include your amended Funding Statement within your cover letter. We will change the online submission form on your behalf.

Reviewers' comments:

Reviewer's Responses to Questions

**Comments to the Author**

1. Is the manuscript technically sound, and do the data support the conclusions?

Reviewer #1: Yes

2. Has the statistical analysis been performed appropriately and rigorously? 

Reviewer #1: Yes

3. Have the authors made all data underlying the findings in their manuscript fully available?

Reviewer #1: Yes

4. Is the manuscript presented in an intelligible fashion and written in standard English?

Reviewer #1: Yes

5. Review Comments to the Author

Reviewer #1: The authors present an interesting study with the aim of identifying cleavage sites leading to the shed forms of human and mouse Klotho protein. It is largely a well written manuscript which is clearly laid out. I have a few minor comments as below:

1. A reference has been missed in the introduction, paragraph 2 "Mice deficient in Klotho exhibit changes observed during human ageing"

2. "They also have a shortened life span"

3. The figure legends are usually at the end of the paper

6. PLOS authors have the option to publish the peer review history of their article (what does this mean?). If published, this will include your full peer review and any attached files.

Reviewer #1: No

---

## [Author Response · Author response to Decision Letter 0]

9 Nov 2019

We are very grateful to the reviewer for the suggestions to improve our manuscript. Our responses are listed below. We hope the new revised manuscript is now suitable for publication in PLOS ONE. Thank you for considering our submission.

Reviewer #1: The authors present an interesting study with the aim of identifying cleavage sites leading to the shed forms of human and mouse Klotho protein. It is largely a well written manuscript which is clearly laid out. I have a few minor comments as below:

1. A reference has been missed in the introduction, paragraph 2 "Mice deficient in Klotho exhibit changes observed during human ageing"

Response: The reference has been added.

2. "They also have a shortened life span"

Response: The word “shortened” has been corrected.

3. The figure legends are usually at the end of the paper

Response: The figure legends are now moved to the end of the paper.

In addition, we provide a document with the entire western blots shown. They were cropped in the manuscript figures.

---

## [Decision Letter · Decision Letter 1]

26 Nov 2019

Identification of the cleavage sites leading to the shed forms of human and mouse anti-aging and cognition-enhancing protein Klotho

PONE-D-19-22466R1

Dear Dr. Abraham,

We are pleased to inform you that your manuscript has been judged scientifically suitable for publication and will be formally accepted for publication once it complies with all outstanding technical requirements.

With kind regards,

Hiroyoshi Ariga

Academic Editor

PLOS ONE

Reviewers' comments:

Reviewer's Responses to Questions

**Comments to the Author**

1. If the authors have adequately addressed your comments raised in a previous round of review and you feel that this manuscript is now acceptable for publication, you may indicate that here to bypass the “Comments to the Author” section, enter your conflict of interest statement in the “Confidential to Editor” section, and submit your "Accept" recommendation.

Reviewer #1: All comments have been addressed

2. Is the manuscript technically sound, and do the data support the conclusions?

Reviewer #1: Yes

3. Has the statistical analysis been performed appropriately and rigorously? 

Reviewer #1: Yes

4. Have the authors made all data underlying the findings in their manuscript fully available?

Reviewer #1: Yes

5. Is the manuscript presented in an intelligible fashion and written in standard English?

Reviewer #1: Yes

6. Review Comments to the Author

Reviewer #1: My comments have been addressed in full. The manuscript is improved and the inclusion of full western blots has been acknowledged.

7. PLOS authors have the option to publish the peer review history of their article (what does this mean?). If published, this will include your full peer review and any attached files.

Reviewer #1: No

---

## [Editor Report · Acceptance letter]

2 Jan 2020

PONE-D-19-22466R1 

Identification of the cleavage sites leading to the shed forms of human and mouse anti-aging and cognition-enhancing protein Klotho 

Dear Dr. Abraham:

I am pleased to inform you that your manuscript has been deemed suitable for publication in PLOS ONE. Congratulations! Your manuscript is now with our production department. 

With kind regards,

on behalf of

Dr. Hiroyoshi Ariga 

Academic Editor

PLOS ONE